# LazyAct: Lazy actor with dynamic state skip based on constrained MDP

**Hongjie Zhang[ID]\*, Zhenyu Chen, Hourui Deng, Chaosheng Feng**

College of Computer Science, Sichuan Normal University, Chengdu, China

\* zhanghongjie@sicnu.edu.cn

**Data Availability Statement:** All relevant data are within the manuscript and its Supporting information files.

**Funding:** This work was supported by the Sichuan Science and Technology Program (No.

## Abstract

Deep reinforcement learning has achieved significant success in complex decision-making tasks. However, the high computational cost of policies based on deep neural networks restricts their practical application. Specifically, each decision made by an agent requires a complete neural network computation, leading to a linear increase in computational cost with the number of interactions and agents. Inspired by human decision-making patterns, which involve reasoning only on critical states in continuous decision-making tasks without considering all states, we introduce the *LazyAct* algorithm. This algorithm significantly reduces the number of inferences while preserving the quality of the policy. Firstly, we incorporate a state skipping branch into the actor network to bypass states with minimal impact. Subsequently, we establish optimization objectives for single-agent and multi-agents inference, incorporating cost constraints based on the IMPALA and MAPPO frameworks, respectively. Finally, we utilize pre-training and fine-tuning techniques to train the policy network. Extensive experimental results indicate that *LazyAct* reduces the number of inferences by approximately 80% and 40% in single-agent and multi-agents scenarios, respectively, while sustaining comparable policy performance. The inferences reduction significantly decreases the time and FLOPs required by the *LazyAct* algorithm to complete tasks. Code is available here https://www.dropbox.com/scl/fo/wyoqo6q9gyt86zobfgbvx/h?\rlkey=0moyxsnoiisfs9y4h89hsou1l&dl=0.

## Introduction

Deep reinforcement learning (DRL) has achieved remarkable success in complex sequential decision-making tasks by integrating neural networks with reinforcement learning techniques. This success spans various domains, including Go [1], autonomous driving [2], and large language models [3, 4], etc. In DRL, agents perceive environmental states and predict optimal actions iteratively to maximize the cumulative reward associated with a given task. Consequently, the computational cost of neural networks scales linearly with the number of interactions between agents and their environment. In multi-agents scenarios, each agent must compute policy decisions for every state, leading to a computational cost that increases proportionally with the number of agents. Reducing the overall inference cost while preserving policy quality is a crucial challenge for the practical deployment of DRL.

2022NSFSC0552). The funders played a role in the study by providing support for decision to publish.

**Competing interests:** The authors have declared that no competing interests exist.

To address the issue of high inference costs, researchers have devised several methods to accelerate prediction processes. These methods can be broadly categorized into two main groups. The first group encompasses neural network compression techniques, including neural network pruning [5–7], weight quantization [8–10], and knowledge distillation [11–13]. The second group involves dynamic neural networks [14], which allocate computing resources adaptively according to the complexity of the data. Examples of such networks include MSDNet [15], which dynamically selects computation branches based on data complexity; GFNet [16], which adjusts image resolution based on local importance; and S2DNAS [17], which automatically searches for optimal dynamic network structures. Dynamic neural networks can accelerate speed even further by integrating these compression techniques. However, it is crucial to note that these solutions still require computation for each sample, and there remains potential for reducing prediction costs in tasks that involve sequential decision-making.

Drawing inspiration from human decision-making, agents can reduce inference costs by not making predictions at every state. We experiment the autonomous driving and SMAC tasks. It illustrates an autonomous driving agent that can repeatedly execute the same action on road segments with no vehicles, thus avoiding the risk of collisions. Similarly, SMAC task presents a scenario of multi-agents cooperative attack, where a specific marine can repeatedly perform the same action without impacting the collective combat. This repetition of actions is analogous to skipping intermediate states. To leverage this characteristic, we introduce the *LazyAct* algorithm in this work, which enhances the original policy network with a state skip branch. This branch allows for the execution of repeated actions in trivial states, bypassing policy computations for these states. This paper draws on the concept of options and **Semi-Markov Decision Processes (semi-MDPs)**, where a sequence of actions over a duration is considered as a high-level action, such as options [18, 19]. The *LazyAct* proposed here is a simplified variant of options, where an option is represented as repeating the first action $k$ times. Moreover, instead of predicting option termination at each state using a terminal function, we predict the duration of the option during action execution. This method reduces the prediction cost of neural networks and simplifies the optimization objective for constrained options. For single agent, our goal is to maximize the cumulative reward of the task while constraining the number of policy computations needed for completion. We formulate the task as a **constrained Markov Decision Process (MDP)** problem to derive the optimization objective and proceed with policy learning accordingly. Specifically, we adapt the constrained MDP for the IMPALA [20] algorithm to handle policy learning in single-agent tasks. In the context of multi-agents systems, we constrain the number of decisions in each state. And we maximize the cumulative reward of the task. In particular, we extend the constrained MDP derivation to the MAPPO [21] algorithm, addressing policy learning for multi-agents tasks. Furthermore, we enhance learning efficiency and policy quality through a combination of unconstrained policy pre-training and constrained policy fine-tuning.

In this work, We have made the following contributions:

1. We have designed the *LazyAct* algorithm tailored for single agent, building upon the IMPALA framework. It is designed to maximize cumulative rewards while enabling it to skip states that are unimportant.

2. For multi-agents scenarios, the *LazyAct* algorithm, grounded in the MAPPO framework, is crafted to maximize the joint reward while respecting constraints on the number of agent decisions per state. To further optimize the training process, we incorporate pre-training and fine-tuning strategies to enhance overall efficiency.

3. Extensive experiments have been conducted in both single-agent and multi-agents tasks. The findings show that *LazyAct* achieves computational savings of 80% in single-agent scenarios and 40% in multi-agents contexts, respectively, while preserving policy performance at a comparable level.

## Related work

### Neural network acceleration

The objective of neural network pruning is to eliminate redundant connections between neurons in the neural network structure. Structured pruning can effectively reduce computational complexity, and researchers have proposed various pruning algorithms targeting convolutional neural networks, including the removal of channels or convolutional kernels [5–7]. Neural network parameter quantization techniques aim to limit the representation accuracy of weights. For instance, compressing 32-bit float parameters into 16-bit float parameters halves the overall network size and achieves doubled computational speed on GPUs. Global quantization selects the minimum numerical precision that maintains accuracy to replace the original values, and further improves accuracy through retraining [22, 23]. DeepCompression [24] employs clustering to divide weights into several clusters, discretizes the weights into several values based on the cluster centers, remarkably reducing the storage space for weights. The motivation behind local quantization is that different layers should adopt parameters with different precisions, such as maintaining high precision for weights closest to the input layer and low precision for weights close to the output layer [25]. Lee et al. proposed an element-wise gradient scaling training algorithm to address the issue of accuracy degradation caused by the direct use of the straight-through estimator (STE) in parameter quantization, aiming to enhance the model's prediction accuracy while maintaining low-precision parameters [26]. Jorn et al. introduced QBitOpt, which can generate mixed-precision networks with high task performance while ensuring strict resource constraints, outperforming fixed-precision methods [27].

In addition to the aforementioned static compression algorithms, dynamic neural networks have emerged in recent years, which aim to reduce inference costs by assigning different computational paths to different samples. Huang et al. proposed the efficient MSDNet, an image classification model that judges the sample difficulty based on the input image's label and selects features from different layers for prediction. When the predicted value of a certain category exceeds a threshold, it is output; otherwise, it proceeds to subsequent feature processing. MSDNet significantly reduces the average prediction cost while maintaining prediction accuracy. To address the challenge of low-level features classification, a multi-scale network structure was designed [15]. Wang et al. discovered that specific regions in images are more critical for recognition tasks. Based on this finding, they proposed the GFNet model, which utilizes reinforcement learning algorithms to lock into important recognition regions for prediction and determines dynamic prediction thresholds based on sample labels. Due to the smaller prediction region, it significantly reduces prediction costs while aligning more closely with human recognition behavior [16]. Cheng et al. leveraged neural architecture search techniques to automatically construct a sample-aware dynamic neural network, which exhibits higher inference efficiency and prediction accuracy compared to manually designed dynamic models [28]. Cui et al. introduced Brainstorm, a deep learning framework for optimizing dynamic neural networks. By unifying the expression of dynamics, Brainstorm fills the gap and through its proposed dynamic optimization, can increase the speed of popular dynamic neural networks by up to 11 times [29].

The accelerated inference idea of this topic is similar to that of dynamic neural networks. Most of the aforementioned dynamic neural networks are aimed at image recognition, and reduce inference costs by controlling resolution, recognition areas, etc. However, they cannot cope with non-image scenarios. In non-image scenarios, this paper refers to the idea of dynamic inference, and only performs inference on critical states to reduce total costs.

### Action repeat technology

There exists a wide range of research areas within RL that focus on addressing the challenge of determining the optimal control frequency, as well as exploring the subject of action repetition. Typically, the motivations behind the repetition of actions vary, such as exploration, improving the signal-to-noise ratio, and managing sample complexity. Despite these diverse reasons, the methodologies employed to tackle these issues share remarkable similarities. These approaches often involve balancing the trade-offs between computational efficiency, accuracy and robustness, aiming to optimize performance by carefully selecting when and how often to repeat certain actions. Alex et al. demonstrate that establishing an appropriate frame skip can be pivotal for the performance of agents trained to play Atari 2600 games [30]. Across all six games, the frame skip proved to be a significant factor in achieving success. Notably, for two of these games, a substantial frame skip resulted in best performance. Adil et al. propose an examination of how the frame skipping rate affects the agent's learning process and ultimate performance, specifically exploring its impact through the application of deep Q-learning, experience replay memory, and the utilization of the ViZDoom Game AI research platform [31]. Alberto et al. introduce the concept of action persistence, which involves repeating an action for a set number of decision steps, thereby altering the control frequency [32]. Initially, they examine how action persistence impacts the efficacy of the optimal policy. Following this, they introduce a new algorithm called PFQI, which is an extension of FQI, aimed at learning the optimal value function for a specific level of persistence. Sharma et al. introduce a unique framework, Fine Grained Action Repetition (FiGAR), that grants the agent the capability to choose both the action and the frequency of its repetition [33]. By facilitating temporal abstractions in the action space, FiGAR can enhance any Deep Reinforcement Learning algorithm that relies on an explicit policy estimate. Andre et al. present a proactive approach where the agent not only picks an action in a given state but also determines the duration of commitment to that action [34]. The method incorporates skip connections between states and learns a skip-policy to repeat the same action across these skips. Sabbioni et al. have devised a unique operator, named the All-Persistence Bellman Operator, which enables efficient utilization of both low-persistence experiences through sub-transition decomposition and high-persistence experiences due to the implementation of an appropriate bootstrap method [35].

The above mentioned action repeat method mainly focuses on single agent and usually aims to maximize cumulative rewards without constraints. In this paper, the purpose of action repeat is to reduce inference costs and ensure the quality of strategies, thereby modeling it as a constrained RL problem. Furthermore, we propose a complete solution for both single-agent and multi-agents scenarios.

## Methodology

### Overview

Fig 1 illustrates the framework of *LazyAct*. Each policy network comprises state skip and action branches, wherein the action output is based on both the skip and the current state.

Initially, the agent determines the skipped length by analyzing the present state. Following this, the policy network utilizes both the skipped length and the current state to identify the

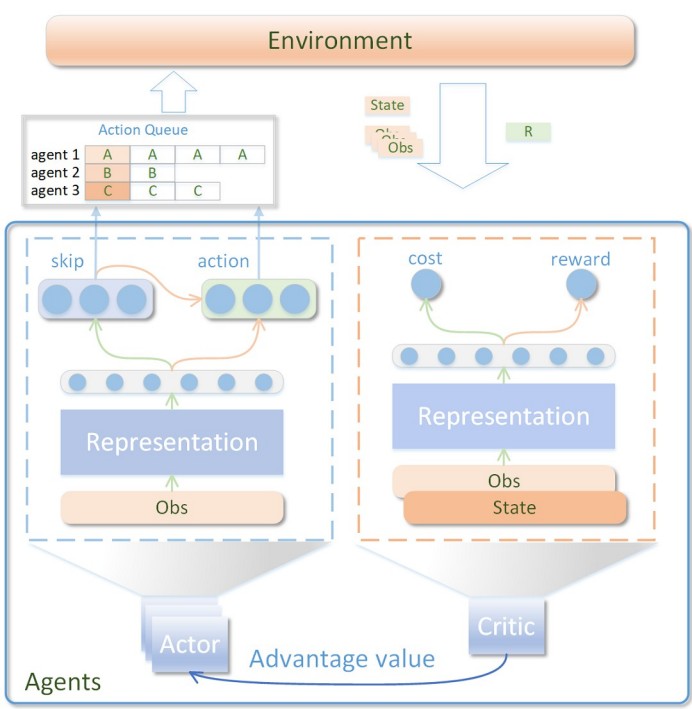

**Fig 1. In the *LazyAct* architecture, a decision is made at each time, the agent returns multiple consecutive actions to achieve state skip.** The actor determines and stores the corresponding actions and skip duration for multi-agents within the environment in the **Action Queue**.

appropriate action. This sequence of actions is then sent to the environment for implementation. Furthermore, the value network integrates a cost value branch and a state value branch. The former evaluates the computational expenses linked to the current state, whereas the latter assesses the overall accumulated reward. The actor determines and stores the corresponding actions and skip duration for multi-agents within the environment in the **Action Queue**. Specifically, when an agent's action queue is depleted, the environment forwards the most recent observation to the agent and computes the upcoming action list. The state, observations and rewards are subsequently employed to refine the policy network, adhering to the principles of the constrained MDP framework.

### *LazyAct* for single-agent based on IMPALA

For single-agent tasks, we devise the *LazyAct* algorithm, which is grounded in the IMPALA framework. IMPALA stands for a distributed deep reinforcement learning training framework, distinguished by its segregation of Actors and Learner roles. Actors engage with their respective environments across multiple servers, relaying experience data to the central Learner. The Learner updates the policy network parameters and disseminates the latest parameters back to the Actor servers. Notably, IMPALA introduces the V-trace algorithm, which addresses off-policy issues and enhances the efficiency of policy learning.

**Formal problem definition.**   We formulate *LazyAct* as a constrained policy optimization problem, as expressed in Eq (1). Here, $\tau$ denotes the interaction trajectory, and $\gamma$, belonging to the interval [0, 1], is the discount factor. $R_t$ signifies the reward at time step $t$. Specifically, $\mathbb{I}(k_t)$ indicates whether to skip the state $s_t$, with $k_t$ denoting the skip decision output. If $\mathbb{I}(k_t) = 1$, it

implies that state $s_t$ is skipped. $\epsilon$ and $\mathbb{N}$ represent the skip constraint and episode length, respectively. The constraint ensures that the average skip ratio across completed tasks is greater than or equal to the predefined value $\epsilon$.

$$max \qquad \mathbb{E}_{\tau \sim p(\tau)} \left[ \sum_{t=0}^{\infty} \gamma^t R_t \right]$$

$$s.t. \qquad \frac{1}{\mathbb{N}} \mathbb{E}_{\tau \sim p(\tau)} \left[ \sum_{t=0}^{\infty} \mathbb{I}(k_t) \right] \geq \epsilon \tag{1}$$

Utilizing Lagrange multipliers, we convert the constrained optimization problem presented in Eq (1) into an unconstrained one, as depicted in Eq (2). Here, $\theta$ and $\alpha$ denote the neural network parameters and the Lagrange multiplier, respectively. $\theta^*$ and $\alpha^*$ correspond to the optimal solutions to the transformed problem.

$$(\theta^*, \alpha^*) = \arg \min_{\alpha \geq 0} \max_{\theta} L(\theta, \alpha)$$

$$where \quad L(\theta, \alpha) = \mathbb{E}_{\tau \sim p(\tau)} \left[ \sum_{t=0}^{\infty} \gamma^t R_t \right] - \alpha \left( \epsilon - \frac{1}{\mathbb{N}} \mathbb{E}_{\tau \sim p(\tau)} \left[ \sum_{t=0}^{\infty} \mathbb{I}(k_t) \right] \right) \tag{2}$$

To simplify the problem, we scale the constraints in Eq (1) with a discount factor $\gamma$, as shown in Eq (3).

$$\epsilon \leq \frac{1}{\mathbb{N}} \mathbb{E}_{\tau \sim p(\tau)} \left[ \sum_{t=0}^{\infty} \gamma^t \mathbb{I}(k_t) \right] \leq \frac{1}{\mathbb{N}} \mathbb{E}_{\tau \sim p(\tau)} \left[ \sum_{t=0}^{\infty} \mathbb{I}(k_t) \right]$$

$$where \quad \forall t, \gamma^t \mathbb{I}(k_t) \leq \mathbb{I}(k_t) \tag{3}$$

We formulate the optimization objective, as detailed in Eq (4), and subsequently divide it into policy and multiplier optimization components. Specifically, the policy and multiplier optimizations are conducted iteratively. In the policy optimization phase, we augment the original reward with skip rewards to incentivize the skipping of certain states. The multiplier optimizer is employed to ensure constraint satisfaction. When the skip ratio meets the $\epsilon$ constraint, $\alpha$ is set to 0; otherwise, $\alpha$ tends towards positive infinity.

$$\arg \min_{\alpha \geq 0} \max_{\theta} \mathbb{E}_{\tau \sim p(\tau)} \left[ \sum_{t=0}^{\infty} \gamma^t (R_t + \alpha \mathbb{I}(k_t)) \right] - \alpha \mathbb{N} \epsilon$$

$$Policy: \qquad \max_{\theta} \mathbb{E}_{\tau \sim p(\tau)} \left[ \sum_{t=0}^{\infty} \gamma^t (R_t + \alpha \mathbb{I}(k_t)) \right] \tag{4}$$

$$Multiplier: \qquad \arg \min_{\alpha \geq 0} - \alpha \left( \epsilon - \frac{1}{\mathbb{N}} \mathbb{E}_{\tau \sim p(\tau)} \left[ \sum_{t=0}^{\infty} \mathbb{I}(k_t) \right] \right)$$

It is equivalent to giving a penalty for not skipping. Because $R_t + \alpha \mathbb{I}(k_t) \propto R_t - \alpha(1 - \mathbb{I}(k_t))$, where the $(1 - \mathbb{I}(k_t))$ can be substituted by penalty term. We use $C_t$ to represent the penalty term, that is, $C_t = 0$ if $\mathbb{I}(k_t) = 1$, otherwise $C_t > 0$. The policy

target is defined as Eq (5).

$$\max_{\theta}\mathbb{E}_{\tau \sim p(\tau)}\left[\sum_{t=0}^{\infty}\gamma^t(R_t - \alpha C_t)\right] \tag{5}$$

***LazyAct* with V-trace.** We extend the policy target in Eq (4) to the version of V-trace, which is the core of IMPALA. The original state value function of V-trace is defined as Eq (6).

$$vs_t = V(s_t) + \sum_{t'=t}^{t+n-1}\gamma^{t'-t}\left(\prod_{i=t}^{t'-1}c_i\right)\delta_t V \tag{6}$$

The V-trace $v_s^t$ is computed recursively, as defined in Eq (7). Here, $\rho_t$ and $c_t$ denote the truncated importance sampling factors. The symbols $\pi$ and $\mu$ refer to the current policy and the actor's policy, respectively, with $\mu$ being a delayed or outdated policy. Additionally, $V(s_t)$ and $C(s_t)$ represent the value function and the cost function, respectively, as illustrated in Fig 1.

$$\rho_t = min\left(\bar{\rho},\frac{\pi(a_t|s_t)}{\mu(a_t|s_t)}\right)$$
$$c_t = min\left(\bar{c},\frac{\pi(a_t|s_t)}{\mu(a_t|s_t)}\right)$$
$$\delta_t V = \rho_t(R_t + \gamma V(s_{t+1}) - V(s_t)) \tag{7}$$
$$vs_t = V(s_t) + \delta_t V + \gamma c_t(vs_{t+1} - V(s_{t+1}))$$
$$\sigma_t C = \rho_t(C_t + \gamma C(s_{t+1}) - C(s_t))$$
$$cs_t = C(s_t) + \sigma_t C + \gamma c_t(cs_{t+1} - C(s_{t+1}))$$

Based on the value estimate of V-trace, we define the policy gradients of skip and action branch as Eq (8).

$$log\pi_{\theta}(a_t, k_t|s_t) = log\pi_{\theta}(k_t|s_t) + log\pi_{\theta}(a_t|k_t, s_t)$$
$$adv(s_t, k_t, a_t) = \rho_t(R_t - \alpha C_t + \gamma(vs_{t+1} - \alpha cs_{t+1}) - (V(s_t) - \alpha C(s_t)) \tag{8}$$
$$pg = -adv(s_t, k_t, a_t)log\pi_{\theta}(a_t, k_t|s_t)$$

**Multiplier optimization.** Drawing from the definition provided in Eq (4), we derive a linear update rule for optimizing $\alpha$, as outlined in Eq (9). Here, $\eta$ represents the learning rate for $\alpha$. This implies that when the skip ratio exceeds the threshold $\epsilon$, $\alpha$ is gradually reduced towards 0, whereas if the skip ratio falls short of $\epsilon$, $\alpha$ is incrementally increased to penalize the insufficient skipping of states.

$$\alpha = max\left(\alpha + \eta\left(\epsilon - \frac{\sum_t\mathbb{I}(k_t)}{\mathbb{N}}\right), 0\right) \tag{9}$$

However, the frequent updates of $\alpha$ can lead to instability in the policy's advantage function *adv*, which in turn affects the convergence of *LazyAct*. To address this issue, we establish an update frequency ratio of 1 : 100 for policy and multiplier updates. This means that $\alpha$ is updated once for every 100 rounds of policy updates, ensuring a more stable learning process.

## *LazyAct* for multi-agents based on MAPPO

In multi-agent systems, planning, consideration and executing are key steps that constitute the process of decision-making. This paper primarily focuses on model-free reinforcement learning. The model-based algorithms to be further investigated in future work. Within the model-free framework, agents lack knowledge of the environment, including rules and rewards, and learn policy through trial and error. Planning involves determining how to achieve complete tasks. Planning can be centralized or decentralized. In centralized planning, a central decision-maker plans for all agents, whereas in decentralized planning, each agent independently creates its own action plan. The baseline algorithm in this work, MAPPO, is an example of decentralized planning. Each agent outputs actions based on observations and receives environmental rewards, continuously adjusting actions to maximize cumulative rewards, ultimately forming the optimal policy $\pi$. The planning, consideration and executing processes is implicitly contained within the policy $\pi$, as $\pi$ has learned the long-term planning for task completion through numerous interactions. *LazyAct* evaluates the importance of states and determines the skip length. *LazyAct* uses a neural network to decide whether to skip states based on their importance. If a state is skipped, *LazyAct* generates a series of repeated actions based on the current state and the skip length.

For multi-agents scenarios, we develop *LazyAct* by building upon the MAPPO algorithm. MAPPO is an extension of the PPO algorithm for multi-agents settings. It employs a framework of centralized learning with decentralized execution. As shown in Fig 2, multiple agents interact independently with the environment and send experiential data to a central learner. The central learner utilizes this data to update the Actor and Critic networks, then sent the updated Actor network back to each agent. After training, the Critic network is no longer used, and only the Actor network used for inference. Our skip is a branch appended to the Actor network. The training of skip is integrated with the original actions, guided by the Critic network during centralized learning. To refine the evaluation of the advantage function, we integrate the Generalized Advantage Estimator (GAE) [36] into Critic. The purpose of *LazyAct* is to reduce the computational cost in MAPPO for each agent by dynamically skipping

**Fig 2. The MAPPO adopts a mode of centralized training with decentralized execution, where multiple agents interact independently with the environment.** The central learner uses experience data to update the Actor and Critic networks and then sends the updated Actor network back to each agent. *LazyAct* outputs skip $k_t$ to skip unimportant states.

unimportant states. This mechanism is compatible with the MAPPO framework and can enhance its performance.

As depicted in Fig 2, the MAPPO process is a decentralized execution paradigm where each agent can only observe its own local observation $o$, without access to the global state $s$ or communication with other agents. Consequently, agents must make decisions based on partially observable $o$. Unlike centralized execution, the action space in MAPPO does not increase with the number of agents. Moreover, the multi-agent system can decompose the complex joint action space into several smaller subspaces, allowing each agent to focus on and execute only a subset of actions, thereby simplifying the decision-making process. Regarding the aspect of skipping, LazyAct can better control the proportion of skips compared to TempoRL, satisfying specific constraints, such as requiring only a certain proportion of agents to make decisions in each state.

**Formal problem definition.** We formulate *LazyAct*, grounded in the MAPPO framework, as a constrained policy optimization problem, as expressed in Eq (10). Here, $o$ denotes the partial observation of an agent, with $j$ indicating the $j$-th agent. $\mathbb{M}$ signifies the number of agents within the multi-agents task. The indicator function $\mathbb{I}(o_j) = 1$ if the $j$-th agent chooses to skip policy calculation at state $s_t$. The constraint in Eq (10) implies that for each state, the proportion of agents not engaging in decision-making should exceed the threshold $\epsilon$, thereby diminishing the computational expense associated with processing each state.

$$max \qquad \mathbb{E}_{(s,o,k,a)\sim\tau_{old}}[min(r(\theta)A(s,o,k,a), clip(r(\theta), 1-\varepsilon, 1+\varepsilon)A(s,o,k,a))]$$

$$s.t. \qquad \mathbb{E}_{(o,k,a)\sim\tau_{old}}\left[\frac{\sum_j \mathbb{I}(o_j)}{\mathbb{M}}\right] \geq \epsilon \tag{10}$$

$$where \qquad r(\theta) = \frac{\pi_\theta(a,k|o)}{\pi_{\theta_{old}}(a,k|o)}$$

In the MAPPO variant of *LazyAct*, the agent predicts both skip and action based on its observation $o$. The computation of the advantage function $A(s, o, k, a)$ takes into account the global state $s$ to facilitate credit assignment among the multiple agents. Analogous to the advantage function in IMPALA, we treat the skip as a cost $C_t$ that is subtracted from the reward $R_t$, and we employ the Lagrange multiplier $\alpha$ to ensure that the constraints are met. Eq (11) delineates the advantage function derived from GAE. Here, $\lambda$ is the GAE parameter that determines the balance between the weight of current returns and future returns. The term $A(s, o, k, a)$ is equivalent to the $gae_t$.

$$dr_t = R_t + \gamma V(s_{t+1}) - V(s_t)$$

$$dc_t = C_t + \gamma C(s_{t+1}) - C(s_t)$$

$$d_t = dr_t + \alpha dc_t \tag{11}$$

$$gae_t = d_t + \gamma\lambda gae_{t+1}$$

The policy definition is presented in Eq (12), encompassing both skip and action branches. Diverging from the single-agent case, the policy $\pi_\theta$ is conditional on the agent's observation $o$. Within our framework, all agents utilize a shared neural network structure and parameter set.

$$log\pi_\theta(a_t, k_t|o_t) = log\pi_\theta(k_t|o_t) + log\pi_\theta(a_t|k_t, o_t) \tag{12}$$

**Multiplier optimization.** The multiplier controls the influence of skip cost. We apply the similar linear updater to $\alpha$, which is shown in Eq (13). Where $\mathbb{M}$ and $\mathbb{N}$ represent the number of agent and episode length, respectively.

$$\alpha = max\left(\alpha + \eta\left(\epsilon - \frac{1}{\mathbb{N}}\sum_t \frac{\sum_j \mathbb{I}(o_j)}{\mathbb{M}}\right), 0\right) \tag{13}$$

## Unconstrained pre-training and constrained fine-tuning

Directly training a policy with skip constraints can result in unstable learning and potential policy collapse, particularly in tasks characterized by sparse rewards. To mitigate this, we employ unconstrained pre-training within the *LazyAct* framework. We initialize the skip ratio threshold $\epsilon$ to 0.0 to facilitate the training of the policy network parameter $\hat{\theta}$, focusing on maximizing cumulative reward. Subsequent to this pre-training phase, we introduce the specified constraints and fine-tune the last layer of the neural network based on the pre-trained parameter $\hat{\theta}$. We then proceed with further training to derive the policy network $\theta$ that adheres to the imposed constraints.

The benefit of this approach is that the agent attains a comprehensive understanding of the environment and establishes an initial policy $\pi$. Leveraging this policy $\pi$, the agent can effectively explore various skip options while maintaining the integrity of the policy. However, if the threshold $\epsilon$ is set too large, it may still lead to policy collapse. In certain tasks, skipping states can prevent the policy from learning valuable information. Consequently, the agent may find that no policy can fulfill the specified constraints. This suggests that different tasks have varying ceilings for $\epsilon$. We ascertain this upper limit by incrementally increasing $\epsilon$. Typically, Our policy learning is performed according to the $\epsilon$ constraint provided by the user. When the policy collapses, it means that the constraints are too strict, and it is challenging to learn a policy that satisfies the $\epsilon$ constraint. In such cases, the user may need to reconsider and relax the constraints to facilitate successful learning.

## Experiments

### Training setups

We have implemented our *LazyAct* algorithm within the frameworks of IMPALA and MAPPO, utilizing PyTorch 1.10.1. All experimental procedures were conducted on a GPU server equipped with a GTX 1080Ti graphics processing unit and an Intel Xeon Gold 5118 processor, which features 48 cores. To mitigate the impact of random seeds and to obtain robust average outcomes, we conducted three replicate experiments for each algorithm. In the context of single-agent tasks, we performed experimental comparisons across 6 Atari tasks. For multi-agents scenarios, we established 5 distinct tasks within the SMAC environment, as detailed in Table 1.

**Table 1. Single agent and multi-agents tasks for *LazyAct*.**

| Agents | Task 1 | Task 2 | Task 3 | Task 4 | Task 5 | Task 6 |
|---|---|---|---|---|---|---|
| Single-Agent | alien | bank heist | beam rider | breakout | pong | riverraid |
| Multi-Agents | 3m | 8m | 25m | 3s2z | 3s5z | - |

**Table 2. Network architectures for single-agent task.** The "Conv" columns show the filer shape of the convolution, Channel(Kernel-size).

| Network | Input | Conv.1 | Conv.2 | Conv.3 | F.C.1 | Skip | Action |
|---|---|---|---|---|---|---|---|
| Actor | 4×84×84 | 32(8) | 64(4) | 64(3) | 512 | 10 | [1–18] |
| Critic | 4×84×84 | 32(8) | 64(4) | 64(3) | 512 | 1 | 1 |

In the above single-agent tasks, the state is image. So we use the convolutional neural network (CNN) as the feature network, and its network structure is shown in Table 2. The "Conv" columns show the filer shape of the convolution, Channel(Kernel-size). We set the skip range is $0 \rightarrow 9$.

In the multi-agents tasks, the state is vector. So we use the fully-connected neural network (FNN) as the feature network, and its network structure is shown in Table 3. And we set the skip range is $0 \rightarrow 2$. Specifically, the input of critic contains observation $o$, global state $s$ and the average current skip ratio.

The compared baseline algorithms are listed as follow:

1. **IMPALA** [20]: It is a parallel Actor-Critic training framework that decouples sampling and training to maximize system throughput, and utilizes value correction to compensate for off-policy issues.

2. **MAPPO** [21]: It is an extension of the PPO algorithm on multi-agents systems and is currently one of the most effective algorithms.

3. **TempoRL** [34]: It provides an evaluation of the skip based on the state and action.

## Scores vs Skip ratios

**The comparison in single-agent tasks.**   We analyze the training curves of *LazyAct*, IMPALA, and TempoRL in terms of cumulative reward, as depicted in Fig 3. Subsequently, we examine the training skip ratio curves of *LazyAct* across various threshold values of $\epsilon$, as illustrated in Fig 4. Additionally, we compare the evolution of the parameter $\alpha$ in *LazyAct* for different threshold values of $\epsilon$, as shown in Fig 5. The findings indicate that *LazyAct* surpasses IMPALA in both sample efficiency and policy quality. We delve into the reasons behind these results. In environments with sparse rewards, the ability to skip allows for the rapid feedback of future rewards to the current state, thereby enhancing reward density. Furthermore,

**Table 3. Network architectures for multi-agents task.**

| Network | Input | F.C.1 | F.C.2 | Skip | Action |
|---|---|---|---|---|---|
| Actor(3m) | 30 | 64 | 64 | 3 | 9 |
| Critic(3m) | 79 | 64 | 64 | 1 | 1 |
| Actor(8m) | 80 | 64 | 64 | 3 | 14 |
| Critic(8m) | 249 | 64 | 64 | 1 | 1 |
| Actor(25m) | 250 | 64 | 64 | 3 | 31 |
| Critic(25m) | 1201 | 64 | 64 | 1 | 1 |
| Actor(2s3z) | 80 | 64 | 64 | 3 | 11 |
| Critic(2s3z) | 201 | 64 | 64 | 1 | 1 |
| Actor(3s5z) | 128 | 64 | 64 | 3 | 14 |
| Critic(3s5z) | 345 | 64 | 64 | 1 | 1 |

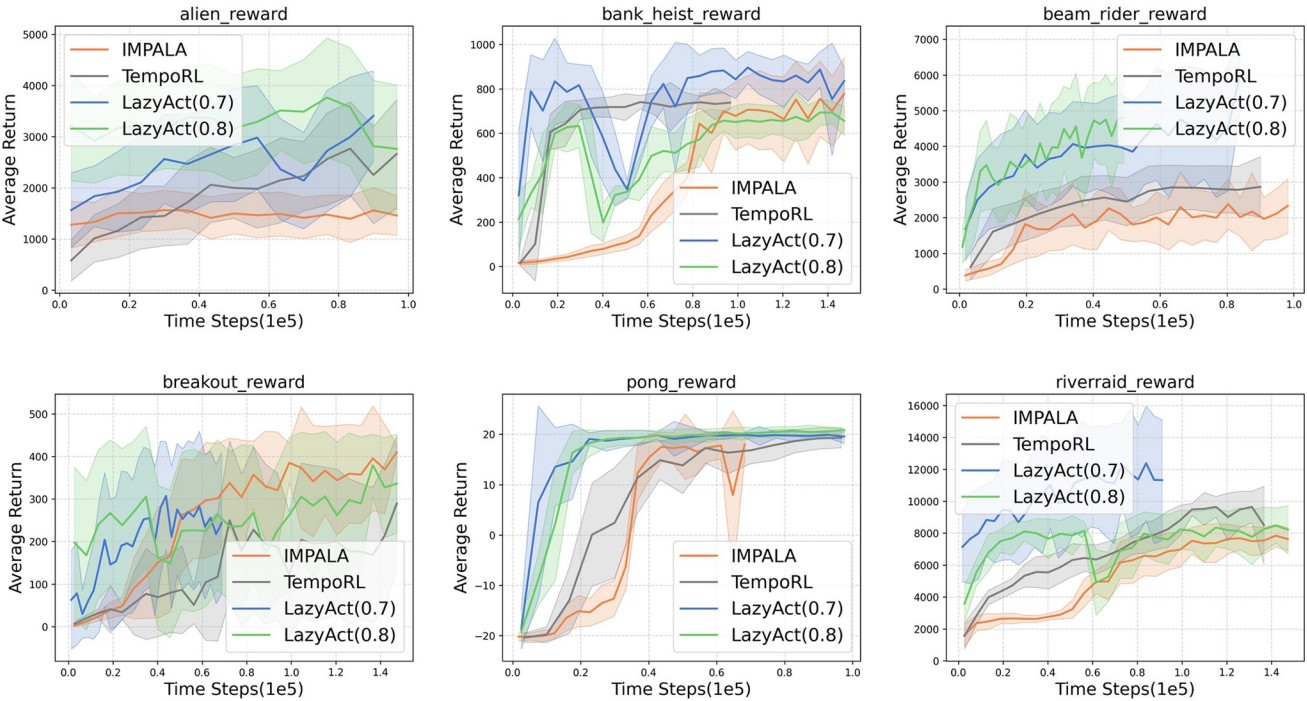

**Fig 3. The score curves of *LazyAct*, IMPALA and TempoRL.** *LazyAct* starts training from an unconstrained pre-trained model.

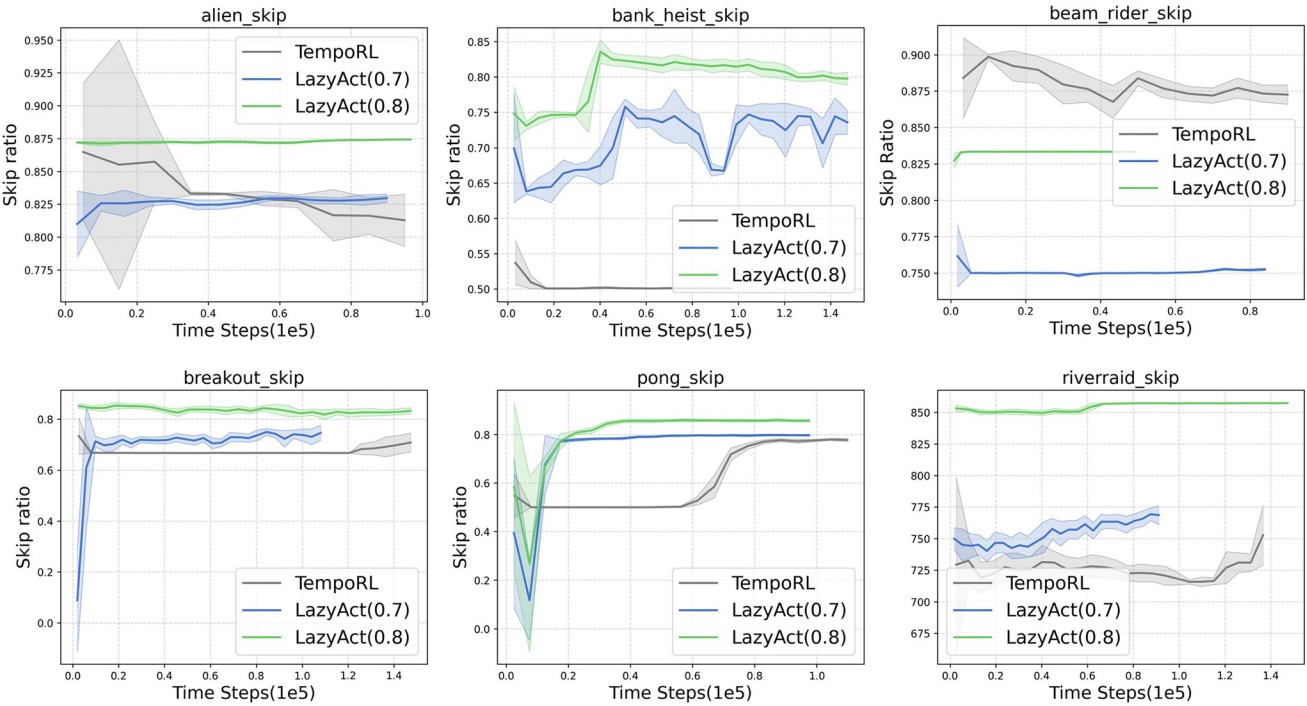

**Fig 4. The skip ratio curves of *LazyAct* with different $\epsilon$ in single-agent tasks.**

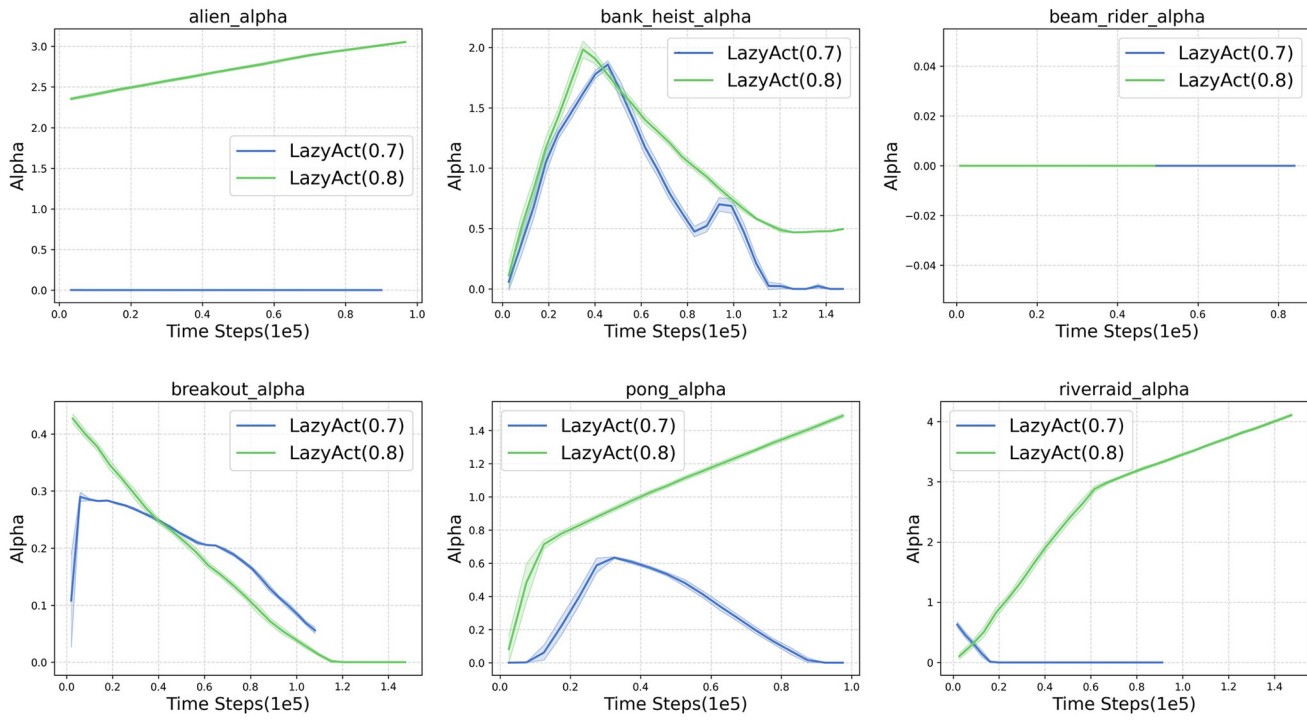

**Fig 5. The alpha($\alpha$) curves of *LazyAct* with different $\epsilon$ in single-agent tasks.**

TempoRL exhibits superior performance to IMPALA in both score and skip metrics. However, TempoRL's lack of constraints necessitates the exploration of a broader search space, which slightly detracts from its score and skip outcomes compared to our *LazyAct* algorithm.

We evaluate the trained policies obtained from *LazyAct*, IMPALA, and TempoRL, examining their cumulative rewards and skip ratios, as presented in Table 4. The values within each cell are denoted as **score (skip ratio)**, with our algorithms $LazyAct_{0.7}$ and $LazyAct_{0.8}$ indicating the outcomes when the skip threshold $\epsilon$ is set to 0.7 and 0.8, respectively. The data reveal that *LazyAct* markedly enhances the skip ratio without compromising the score; in fact, its average score surpasses that of both IMPALA and TempoRL. This can be explained by the principle that skipping irrelevant states accelerates the feedback of future rewards to the current state *s*, particularly in tasks with sparse rewards. Moreover, the skip ratio achieved by *LazyAct* aligns with the specified threshold $\epsilon$. Additionally, TempoRL is not suitable for tasks that require a predetermined skip ratio due to its lack of control over the skip ratio.

**The comparison in multi-agents tasks.** Given that TempoRL is tailored for single-agent settings and is not directly applicable to multi-agents tasks, we restrict our comparisons to the most robust baseline, MAPPO. We assess the training curves of *LazyAct* and MAPPO in terms

**Table 4. Scores vs Skip ratios on single-agent task.** Each cell represents the score(skip ratio).

|  | alien | bank heist | beam rider | breakout | pong | riverraid |
|---|---|---|---|---|---|---|
| IMPALA [20] | 1576(-) | 707(-) | 2146(-) | 386(-) | 20.1(-) | 8065(-) |
| TempoRL [34] | 2747(0.81) | 746(0.50) | 3064(0.87) | 283(0.70) | 20.2(0.78) | 9676(0.75) |
| $LazyAct_{0.7}$ | **3678**(0.82) | **894**(0.74) | 4533(0.75) | 340(0.74) | 20.0(0.80) | **12553**(0.77) |
| $LazyAct_{0.8}$ | 3034(0.87) | 699(0.80) | **4663**(0.83) | **387**(0.82) | **20.9**(0.84) | 8792(0.85) |

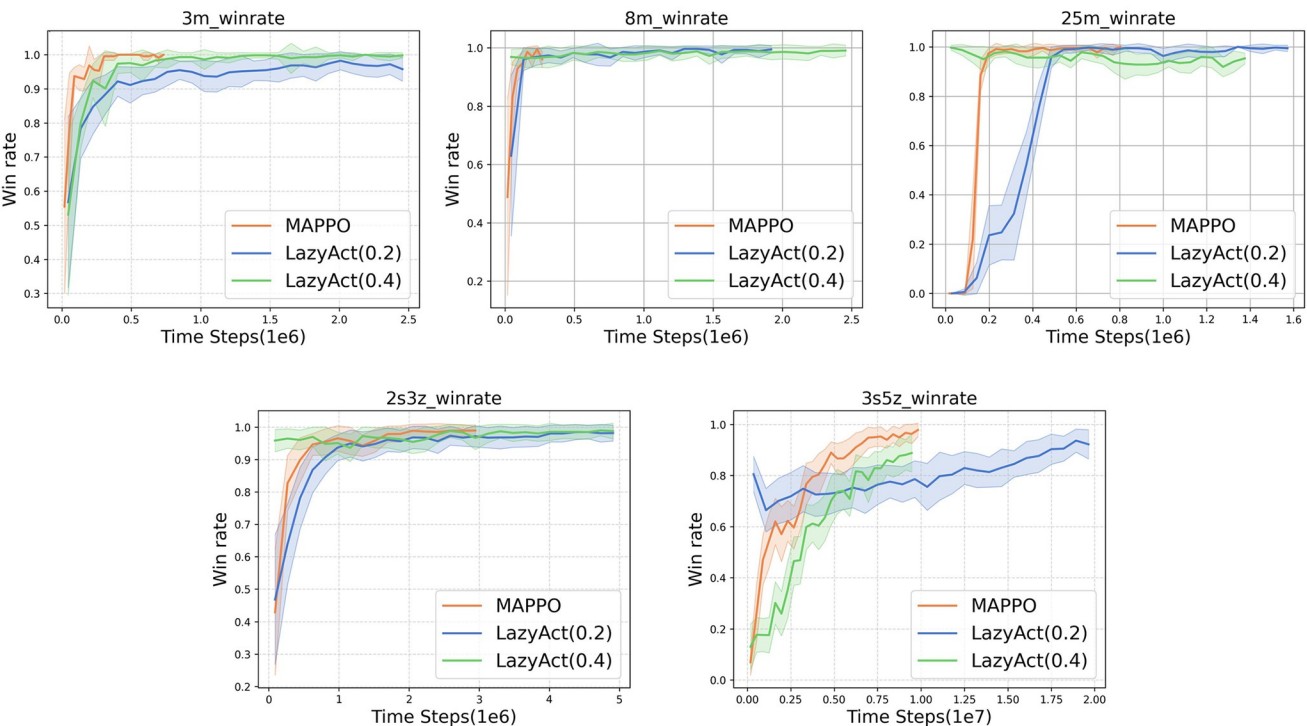

**Fig 6. The win rate curves of *LazyAct* and MAPPO.** *LazyAct* starts training from an unconstrained pre-trained model.

of win rate, as depicted in Fig 6. Subsequently, we examine the training skip ratio curves of *LazyAct* across various threshold values of $\epsilon$, as illustrated in Fig 7. Furthermore, we compare the evolution of the parameter $\alpha$ in *LazyAct* for different threshold values of $\epsilon$, as shown in Fig 8.

We benchmark the trained policies of *LazyAct* against those of MAPPO, evaluating their win rates and skip ratios, as detailed in Table 5. The values within each cell are presented as win **rate (skip ratio)**, with our algorithms *LazyAct*$_{0.2}$ and *LazyAct*$_{0.4}$ indicating the outcomes when the skip threshold $\epsilon$ is set to 0.2 and 0.4, respectively. The data demonstrate that *LazyAct* successfully increases the skip ratio, satisfying the specified constraint threshold $\epsilon$. Concurrently, *LazyAct* maintains a final win rate comparable to that of MAPPO. With a skip ratio of 0.2, the average win rate decrement does not exceed 2%, and when the skip ratio is 0.4, the average win rate decrement is no more than 2.7%.

## Time and FLOPs saving

In this section, we compared the reduction in time and floating-point operations (FLOPs) of LazyAct against the baseline algorithm. Notably, different algorithms require varying steps to complete tasks, and the computation time and FLOPs are not directly proportional to the skip ratio. Moreover, the added skip branches result in extra computation time and FLOPs. Additionally, we measured computation time on the 1080Ti, where due to parallel computing, the time is not directly proportional to FLOPs. So, We directly measured the actual runtime of different algorithms.

Table 6 shows the savings in time and FLOPs of LazyAct compared to baseline algorithms in single-agent tasks. The results indicate that LazyAct significantly reduces both time and

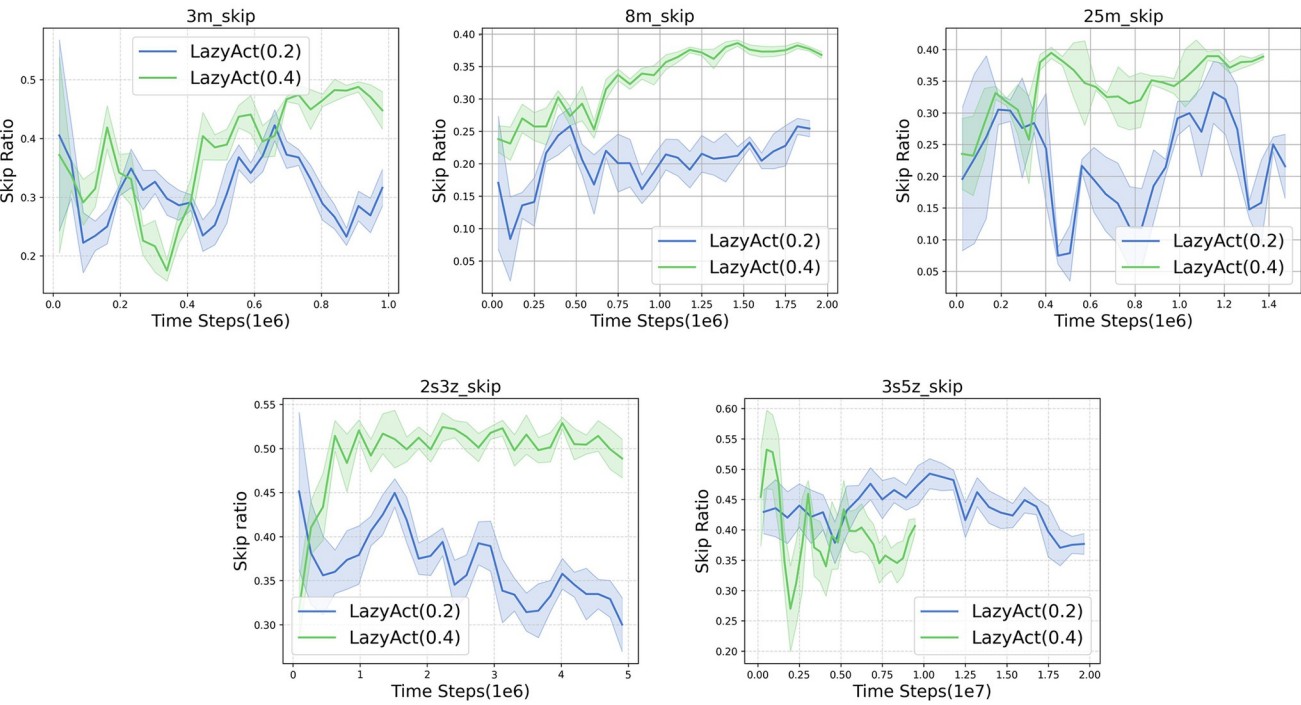

**Fig 7. The skip ratio curves of *LazyAct* with different $\epsilon$ in multi-agents tasks.**

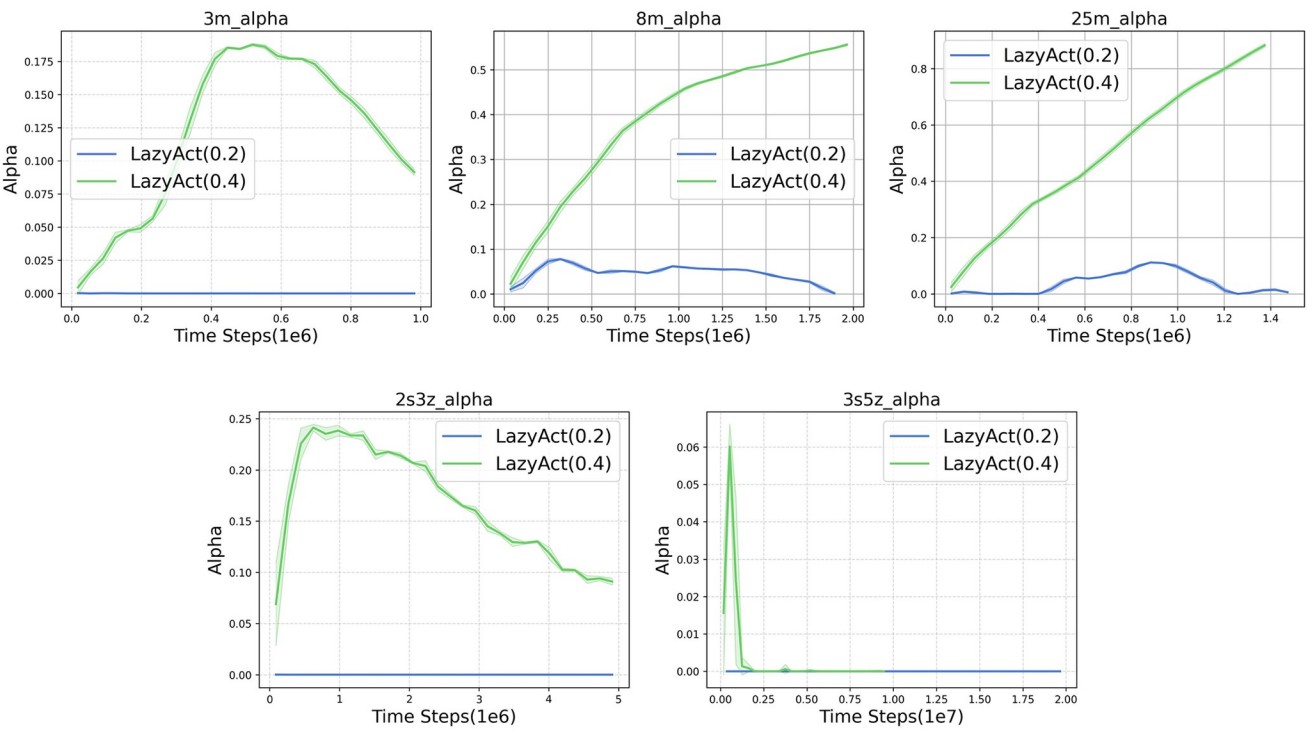

**Fig 8. The alpha($\alpha$) curves of *LazyAct* with different $\epsilon$ in multi-agents tasks.**

**Table 5. Win rate vs Skip ratios on multi-agents tasks.** Each cell represents the Win rate(skip ratio).

|  | 3m | 8m | 25m | 2s3z | 3s5z |
|---|---|---|---|---|---|
| MAPPO [21] | **1.00**(0.00) | **0.98**(0.00) | **1.00**(0.00) | 0.97(0.00) | **0.96**(0.00) |
| $LazyAct_{0.2}$ | 0.90(0.32) | **0.98**(0.25) | **1.00**(0.22) | **0.98**(0.31) | 0.95(0.40) |
| $LazyAct_{0.4}$ | **1.00**(0.45) | 0.96(0.39) | 0.96(0.40) | 0.96(0.51) | 0.90(0.42) |

**Table 6. Time(s) and GFLOPs savings of single-agent tasks.** Each cell represents the Time(GFLOPs).

|  | alien | bank heist | beam rider | breakout | pong | riverraid |
|---|---|---|---|---|---|---|
| IMPALA [20] | 5.00(9.35) | 7.00(13.09) | 10.00(18.70) | 12.50(23.37) | 12.80(23.93) | 5.45(10.19) |
| TempoRL [34] | 1.38(2.21) | 3.36(5.62) | 3.40(3.65) | 3.69(6.18) | 2.59(4.32) | 1.96(3.28) |
| $LazyAct_{0.7}$ | 1.20(2.02) | 2.15(3.60) | 4.76(7.95) | 3.49(5.84) | 2.35(3.93) | 1.93(3.23) |
| $LazyAct_{0.8}$ | **1.09(1.82)** | **1.58(2.65)** | **3.33(5.57)** | **2.01(3.37)** | **1.79(2.99)** | **0.92(1.54)** |

**Table 7. Time(ms) and KFLOPs savings of multi-agents tasks.** Each cell represents the Time(KFLOPs).

|  | 3m | 8m | 25m | 2s3z | 3s5z |
|---|---|---|---|---|---|
| MAPPO [21] | 2.3(118) | 3.0(235) | 4.3(669) | 4.4(348) | 4.8(496) |
| $LazyAct_{0.2}$ | 2.1(88) | 3.0(193) | 4.0(509) | 4.3(259) | 4.0(320) |
| $LazyAct_{0.4}$ | **1.7(71)** | **2.5(157)** | **2.9(361)** | **3.0(183)** | **3.9(309)** |

FLOPs compared to IMPALA and TempoRL, while maintaining high scores. With a higher skip ratio than TempoRL, LazyAct achieves noticeable reductions in time and FLOPs.

Table 7 shows the savings in time and FLOPs of LazyAct compared to baseline algorithms in multi-agents tasks. The results indicate that LazyAct significantly reduces both time and FLOPs compared to MAPPO, while maintaining high scores.

## Toy example

To gain an intuitive understanding of *LazyAct*'s decision-making process, we visualized the behavior of SMAC-25m in a multi-agents setting. Fig 9 illustrates the number of agents making decisions at each state, with gray cells indicating that the agent is running policy inference and white cells signifying a skip decision. In this particular task, *LazyAct* employs only approximately half of the agents to compute the policy.

## Conclusion

This paper introduces *LazyAct*, a novel lazy actor approach. Drawing inspiration from human decision-making processes, we aim to reduce the computational overhead of policy evaluation

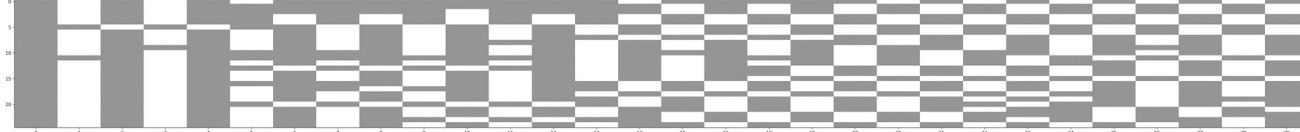

**Fig 9. Visualization on SMAC-25m based on *LazyAct*.**

while preserving policy quality by bypassing irrelevant states. To achieve this, we incorporate a skip branch into the actor network, which is responsible for predicting whether to skip certain states. We have formulated an optimization objective that includes skip mechanisms for both the single-agent algorithm IMPALA and the multi-agents algorithm MAPPO, achieving substantial results on Atari and SMAC tasks, respectively. By capitalizing on the redundancy inherent in sequential decision-making, we dynamically skip states to enhance efficiency. In future research, we plan to adapt *LazyAct* to additional DRL algorithms, making it a versatile and widely applicable module. Moreover, we intend to integrate it with other neural network architectures, such as Transformer, and to develop new optimization techniques.

## Supporting information

**S1 File. The source code of LazyAct.**
(ZIP)

**S2 File. The data of LazyAct.** In reinforcement learning tasks, data is generated from the environment code.
(ZIP)

## Author Contributions

**Conceptualization:** Hongjie Zhang.

**Data curation:** Hourui Deng.

**Formal analysis:** Hongjie Zhang, Zhenyu Chen, Hourui Deng, Chaosheng Feng.

**Funding acquisition:** Hongjie Zhang, Chaosheng Feng.

**Methodology:** Hongjie Zhang, Chaosheng Feng.

**Software:** Hongjie Zhang, Zhenyu Chen, Hourui Deng, Chaosheng Feng.

**Validation:** Hongjie Zhang, Zhenyu Chen, Hourui Deng, Chaosheng Feng.

**Visualization:** Zhenyu Chen, Hourui Deng.

**Writing – original draft:** Hongjie Zhang.

**Writing – review & editing:** Hongjie Zhang, Zhenyu Chen, Chaosheng Feng.

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
