## [Decision Letter · Decision Letter 0]

25 Sep 2024

PONE-D-24-26037LazyAct: Lazy Actor with Dynamic State Skip based on Constrained MDPPLOS ONE

Dear Dr. Zhang,

Thank you for submitting your manuscript to PLOS ONE. After careful consideration, we feel that it has merit but does not fully meet PLOS ONE’s publication criteria as it currently stands. Therefore, we invite you to submit a revised version of the manuscript that addresses the points raised during the review process.

We look forward to receiving your revised manuscript.

Kind regards,

Manoharan Premkumar

Academic Editor

PLOS ONE

“This work was supported by the Sichuan Science and Technology Program (No. 2022NSFSC0552).”

“This work was supported by the Sichuan Science and Technology Program (No. 2022NSFSC0552).”

“This work was supported by the Sichuan Science and Technology Program (No. 2022NSFSC0552).”

5. We note that Figures 1 and 9 in your submission contain copyrighted images. All PLOS content is published under the Creative Commons Attribution License (CC BY 4.0), which means that the manuscript, images, and Supporting Information files will be freely available online, and any third party is permitted to access, download, copy, distribute, and use these materials in any way, even commercially, with proper attribution. For more information, see our copyright guidelines: http://journals.plos.org/plosone/s/licenses-and-copyright.

1. You may seek permission from the original copyright holder of Figures 1 and 9 to publish the content specifically under the CC BY 4.0 license.

Additional Editor Comments:

Address the points raised by the reviewers. Give proper response and action.

Reviewers' comments:

Reviewer's Responses to Questions

**Comments to the Author**

1. Is the manuscript technically sound, and do the data support the conclusions?

Reviewer #1: Yes

Reviewer #2: Yes

2. Has the statistical analysis been performed appropriately and rigorously? 

Reviewer #1: Yes

Reviewer #2: Yes

3. Have the authors made all data underlying the findings in their manuscript fully available?

Reviewer #1: Yes

Reviewer #2: Yes

4. Is the manuscript presented in an intelligible fashion and written in standard English?

Reviewer #1: Yes

Reviewer #2: No

5. Review Comments to the Author

Reviewer #1: General comments

- The manuscript introduced an novel algorithm namely LazyAct, which claimed to significantly reduce the number of inferences while preserving the quality of policy. Theoretically the method proposed in the manuscript demonstrated feasibility of inference reducing, however, there are still a few things need to improve before the manuscript being accepted.

In detail comments

- Normally, a multi-agent system to deal with a complex task chain such as game playing, will include a consideration process such as planning, consideration, executing etc. These seems not been mentioned in the manuscript.

- Regarding to Eq (4), not sure why you provide argmin prior to max, since it seems a backward computation.

- With MAPPO scheme included, authors’ written down “employs a framework of centralized learning with decentrallized execution ”, it seem like another “pull-push” process, cannot understand why you do this repeat and reverse though, please make additional discussion about neccessity.

- Similar as prior question, once using MAPPO framework for executing-reflexion, the purpose to feedback providing referring to partial observation need to discuss, particularly outperforming among other methods. For now, I could not imagine other outperform arpart from computing deduction.

- With concluding remarks in abstract, “Lazyact reduces the number of inferences by approximately 80% and 40%” , however results does not seems to match the justification, as I have not yet seen any discussion regarding to time efficiency or computational infrastructure saving etc.

Reviewer #2: Dear Authors

I would like to congratulate you on a very good approach. From my view, it is a valuable addition to the literature. The only minor revision required is to improve the English language of the manuscript.

6. PLOS authors have the option to publish the peer review history of their article (what does this mean?). If published, this will include your full peer review and any attached files.

Reviewer #1: No

Reviewer #2: **Yes: **Ghassan Abdul-Majeed

---

## [Author Response · Author response to Decision Letter 0]

1 Nov 2024

Dear academic editor and reviewers,

Thank you for providing the opportunity to revise our manuscript titled "LazyAct: Lazy Actor with Dynamic State Skip based on Constrained MDP" (Manuscript ID: PONE-D-24-26037). We appreciate the valuable feedback from the academic editor and reviewers, which has helped us improve the quality of our work. Below, we address each point raised and provide a detailed rebuttal or explanation for the changes made.

Addressing the Comments of the Academic Editor:

Response: We acknowledge the importance of adhering to the journal’s style requirements and file naming conventions. We have reviewed the PLOS ONE author guidelines and have made the necessary adjustments to ensure our manuscript complies with the specified standards.

2. Please note that PLOS ONE has specific guidelines on code sharing for submissions in which author-generated code underpins the findings in the manuscript.

Response: We are confident that these actions fully comply with PLOS ONE’s guidelines on code sharing. We are committed to open science practices and believe that sharing our code will contribute to the advancement of the field.

3. Please include this amended Role of Funder statement in your cover letter.

Response: We have included the amended Role of Funder statement in our cover letter as requested.

4. Please remove any funding-related text from the manuscript and let us know how you would like to update your Funding Statement.

Response: We have removed all funding-related text from the manuscript.

5. We note that Figures 1 and 9 in your submission contain copyrighted images.

Response: We have removed the Figure 1 and revised Figure 9, deleting the copyrighted images.

6. Please include captions for your Supporting Information files at the end of your manuscript.

Response: Captions for the Supporting Information files have been added to the 'Supporting information' of the manuscript.

Addressing the Comments of Reviewer 1:

1. Normally, a multi-agent system to deal with a complex task chain such as game playing, will include a consideration process such as planning, consideration, executing etc. These seems not been mentioned in the manuscript.

Response: Thank you for the suggestion. You pointed out that our paper did not address the consideration process of multi-agent systems when facing complex task chains, such as planning, thinking, and executing. We have supplemented this content in our paper "LazyAct for Multi-agents based on MAPPO" and explained how LazyAct integrates with multi-agent systems.

In multi-agent systems, planning, consideration and execution are key steps that constitute the process of decision-making. Our paper focuses on model-free reinforcement learning, while model-based algorithms will be further studied in future work. In the model-free methods, agents have no knowledge of the environment, including rules and rewards. Agents learn policy through trial and error. Planning involves determining how to achieve goals or complete tasks. Planning can be centralized or decentralized. In centralized planning, there is a central decision-maker for all agents. In decentralized planning, each agent independently makes its own action plan. The MAPPO is a form of decentralized planning. Each agent outputs actions based on its observations and receives environmental rewards, adjusting actions to maximize cumulative rewards. Processes like planning are implicitly included in the policy π, as π has already learned the long-term planning for task completion through numerous interactions. LazyAct evaluates the importance of states and determines the skip length. LazyAct uses a neural network to evaluate the importance of states and decides whether to skip them based on this importance. If a state is skipped, LazyAct generates a series of repeated actions based on the current state and the skip length. LazyAct can be combined with other consideration processes, such as planning and thinking. For example, LazyAct can be used in the planning phase to reduce the number of states that need consideration, thus improving planning efficiency.

2. Regarding to Eq (4), not sure why you provide argmin prior to max, since it seems a backward computation.

Response: Thank you for the suggestion. We employ the Primal Dual Optimization (PDO) method in constrained reinforcement learning. By using Lagrange multipliers, we transform the constrained problem into an unconstrained min-max optimization problem, where 'α' is the Lagrange multiplier. Intuitively, 'α' balances the cumulative reward gained by the agent and the cumulative computational cost. When 'α' is 0, it means the computational cost is not considered. As 'α' increases, the learned policy tends to reduce the computational cost. In Equation 4, the inner max operation represents maximizing cumulative reward given 'α', which considers the computational cost. The outer argmin operation checks if the current policy meets the constraints. If the constraints are met, 'α' is set to 0, prompting the inner layer to maximize cumulative reward. If the constraints are violated, 'α' increases, steering the inner policy towards reducing computational cost to ensure the policy meets the constraints. Through alternating training of the inner and outer layers, It converges to the optimal 'α' that satisfies the constraints, along with the corresponding policy π.

3. With MAPPO scheme included, authors’ written down “employs a framework of centralized learning with decentrallized execution ”, it seem like another “pull-push” process, cannot understand why you do this repeat and reverse though, please make additional discussion about neccessity.

Response: Thank you for the suggestion. You pointed out the "centralized learning, decentralized execution" framework of the MAPPO algorithm and the role of LazyAct within this framework. Our explanation of the MAPPO framework in the paper was not clear enough, and there may have been misunderstandings about the role of LazyAct. We provide additional clarification here:

The MAPPO algorithm uses the "centralized learning, decentralized execution" framework, as shown in Figure 3 of the paper "LazyAct for Multi-agents based on MAPPO."

Multiple agents interact independently with the environment and send experience data to a central learner. The central learner updates the Actor and Critic networks with this data and sends the updated Actor network back to each agent. After training, the Critic network is no longer used, and only the Actor network interacts with the environment. Our 'skip' is a branch inserted into the Actor network, used to evaluate whether subsequent states can adopt repeated actions, thus saving computation for the agents. The 'skip' actions are integrated with the original actions during training, guided by the Critic network. Regarding the "repeat and reverse" process you mentioned, we understand your confusion. In fact, LazyAct does not use a "repeat and reverse" process. The core mechanism of LazyAct is to let each agent in MAPPO reduce the computational cost of policy evaluation by dynamically skipping unimportant states. This mechanism is compatible with the MAPPO framework and can help enhance its performance. We have revised the paper "LazyAct for Multi-agents based on MAPPO" according to your comments and provided a more detailed explanation of the MAPPO framework and the role of LazyAct. We believe that LazyAct can be effectively integrated into the MAPPO framework and help improve the efficiency of multi-agent systems.

4. Similar as prior question, once using MAPPO framework for executing-reflexion, the purpose to feedback providing referring to partial observation need to discuss, particularly outperforming among other methods. For now, I could not imagine other outperform arpart from computing deduction.

Response: Thank you for the suggestion. As described in Figure 3 of "LazyAct for Multi-agents based on MAPPO," the MAPPO process is a decentralized execution model. Each agent can only observe its local observations o, without access to the global state s or communication with other agents. Therefore, agents make decisions based on partially observable states. Unlike a centralized execution model, the action space in MAPPO does not increase with the number of agents. Moreover, the multi-agent system can decompose the complex joint action space into several smaller subspaces. This allows each agent to focus on and execute only a part of the actions, simplifying the decision-making process. Regarding the 'skip', compared to TempoRL, LazyAct can better control the proportion of skips to meet constraints, such as requiring only a certain percentage of agents to make decisions in each state. We have presented experimental results of LazyAct in partially observable environments in the paper, such as win rates and skip proportions in SMAC tasks. The results show that LazyAct can significantly reduce computational costs without sacrificing policy quality and meet specific constraint.

5. With concluding remarks in abstract, “Lazyact reduces the number of inferences by approximately 80% and 40%” , however results does not seems to match the justification, as I have not yet seen any discussion regarding to time efficiency or computational infrastructure saving etc.

Response: Thank you for the suggestion. You correctly pointed out that the paper did not fully discuss the time and computational savings of LazyAct, which is significant. There was indeed a shortfall in the paper. To better illustrate the advantages of LazyAct, we have supplemented the experimental section with results on time and computational savings (FLOPs). The results show that LazyAct significantly reduces the number of inferences while maintaining policy performance, and substantially saves time and FLOPs. The revised version of the paper includes a detailed explanation in the "Time and FLOPs Saving" section.

We have revised the abstract and discussion sections based on the additional experimental results to more accurately reflect the benefits of LazyAct. We believe that the time and computational savings offered by LazyAct increase its practical value in real-world applications.

Addressing the Comments of Reviewer 2:

1. I would like to congratulate you on a very good approach. From my view, it is a valuable addition to the literature. The only minor revision required is to improve the English language of the manuscript.

Thank you for the feedback. We will address the minor revision by improving the English language throughout the manuscript. We will have the manuscript professionally edited to ensure clarity and grammatical accuracy. We highlight changes in ‘Revised Manuscript with Track Changes’ made to the original version. 

The following lists the main revisions made to the paper:

1) In abstraction: However, the high computational cost of policies based on deep neural networks restricts their practical application.

2) Line 5: and large language models [3, 4], etc.

3) Line 22: Dynamic neural networks can accelerate speed

4) Line 39: a duration is considered as a high-level action

5) Line 48: to derive the optimization objective

6) Line 50-52: In the context of multi-agents systems, we constrain the number of decisions in each state. And we maximize the cumulative reward of the task. In particular,

7) Line 58-59: It is designed to maximize cumulative rewards while enabling it to skip states that are unimportant.

8) Line 81: cluster centers, remarkably reducing the storage

9) Line 102: learning algorithms to lock into important recognition

10) Line 110: Brainstorm fills the gap and through its proposed dynamic optimization

11) Line 126: between computational efficiency, accuracy and robustness, 

12) Line 151: The above mentioned action repeat method mainly focuses on single agent 

13) Line 163: This sequence of actions is then sent to the

14) Line 173: LazyAct for Single-agent based on IMPALA

15) Line 175: IMPALA stands for a distributed

16) Line 178: The Learner updates the policy network parameters

17) Line 182: Formal Problem Definition

18) Line 218: Drawing from the definition provided

19) Line 228: LazyAct for Multi-agents based on MAPPO

20) Line 288: agent’s observation o

21) Line 294: Unconstrained Pre-training and Constrained Fine-tuning

22) Line 306: is set too large

23) Line 316: Training Setups

24) Line 339: It provides an evaluation of the skip based on the state and action.

25) Line 368: applicable to multi-agents tasks

In conclusion, we have carefully addressed all the comments and suggestions provided by the academic editor and reviewers. We believe that the revised manuscript is now significantly improved and better aligned with the journal's standards. We hope that the manuscript is now suitable for publication.

Thank you again for your valuable feedback and the opportunity to revise our work.

Sincerely,

Hongjie Zhang

College of Computer Science, Sichuan Normal University, Chengdu, China

zhanghongjie@sicnu.edu.cn

---

## [Decision Letter · Decision Letter 1]

22 Jan 2025

LazyAct: Lazy Actor with Dynamic State Skip based on Constrained MDP

PONE-D-24-26037R1

Dear Dr. Zhang,

We’re pleased to inform you that your manuscript has been judged scientifically suitable for publication and will be formally accepted for publication once it meets all outstanding technical requirements.

Kind regards,

Manoharan Premkumar

Academic Editor

PLOS ONE

Additional Editor Comments (optional):

Reviewers' comments:

Reviewer's Responses to Questions

**Comments to the Author**

1. If the authors have adequately addressed your comments raised in a previous round of review and you feel that this manuscript is now acceptable for publication, you may indicate that here to bypass the “Comments to the Author” section, enter your conflict of interest statement in the “Confidential to Editor” section, and submit your "Accept" recommendation.

Reviewer #1: All comments have been addressed

2. Is the manuscript technically sound, and do the data support the conclusions?

Reviewer #1: Yes

3. Has the statistical analysis been performed appropriately and rigorously? 

Reviewer #1: Yes

4. Have the authors made all data underlying the findings in their manuscript fully available?

Reviewer #1: Yes

5. Is the manuscript presented in an intelligible fashion and written in standard English?

Reviewer #1: Yes

6. Review Comments to the Author

Reviewer #1: General comments

The manuscript could be accepted since all my recommendations been addressed.

In detail comments

- Normally, a multi-agent system to deal with a complex task chain such as game playing, will include a consideration process such as planning, consideration, executing etc. These seems not been mentioned in the manuscript. [√]

- Regarding to Eq (4), not sure why you provide argmin prior to max, since it seems a backward computation. 【√】

- With MAPPO scheme included, authors’ written down “employs a framework of centralized learning with decentrallized execution ”, it seem like another “pull-push” process, cannot understand why you do this repeat and reverse though, please make additional discussion about neccessity.【√】

- Similar as prior question, once using MAPPO framework for executing-reflexion, the purpose to feedback providing referring to partial observation need to discuss, particularly outperforming among other methods. For now, I could not imagine other outperform arpart from computing deduction. 【√】

- With concluding remarks in abstract, “Lazyact reduces the number of inferences by approximately 80% and 40%” , however results does not seems to match the justification, as I have not yet seen any discussion regarding to time efficiency or computational infrastructure saving etc. 【√】

7. PLOS authors have the option to publish the peer review history of their article (what does this mean?). If published, this will include your full peer review and any attached files.

Reviewer #1: No

---

## [Editor Report · Acceptance letter]

28 Jan 2025

PONE-D-24-26037R1 

PLOS ONE

Dear Dr. Zhang, 

I'm pleased to inform you that your manuscript has been deemed suitable for publication in PLOS ONE. Congratulations! Your manuscript is now being handed over to our production team.

Kind regards, 

on behalf of

Dr. Manoharan Premkumar 

Academic Editor

PLOS ONE